# Dynamics of Lipid Profile in Antiretroviral-Naïve HIV-Infected Patients, Treated with TAF-Based Regimens: A Multicenter Observational Study

**DOI:** 10.3390/biomedicines10123164

**Published:** 2022-12-07

**Authors:** Salvatore Martini, Paolo Maggi, Cristina Gervasoni, Lorenzo Onorato, Sergio Ferrara, Loredana Alessio, Chiara Bellacosa, Vincenzo Esposito, Giovanni Di Filippo, Addolorata Masiello, Adelaide Maddaloni, Simona Madonia, Giovanna D’Alessio, Viviana Rizzo, Nicola Coppola

**Affiliations:** 1Department of Mental Health and Public Medicine, Section of Infectious Diseases, University of Campania, Luigi Vanvitelli, 80131 Naples, Italy; 2AORN Sant’Anna e San Sebastiano of Caserta, 81100 Caserta, Italy; 3Department of Infectious Diseases, ASST Fatebenefratelli Sacco University Hospital, 20157 Milan, Italy; 4Department of Medical and Surgical Sciences, Section of Infectious Diseases, University of Studies of Foggia, 71122 Foggia, Italy; 5Department of Precision and Regenerative Medicine and Jonian Area, Section of Infectious Diseases, University of Studies of Bari, 70121 Bari, Italy; 6IV Division AORN dei Colli, 80131 Naples, Italy; 7Department of Medicine and Surgery, Section of Infectious Diseases, University Federico II of Naples, 80138 Napoli, Italy; 8AO San Giuseppe Moscati of Avellino, 83100 Avellino, Italy; 9VII Division AORN dei Colli, 80131 Napoli, Italy; 10AOU San Giovanni di Dio e Ruggi d’Aragona of Salerno, 84125 Salerno, Italy; 11AORN San Pio of Benevento, 82100 Benevento, Italy

**Keywords:** HIV, hypercholesterolemia, TAF, TDF, dyslipidemia, cardiovascular risk

## Abstract

**Background:** The introduction of tenofovir alafenamide (TAF) in antiretroviral therapy has deeply modified the choice of the backbone for different treatment regimens, allowing the prevention of the bone and renal toxicity that was related to the previous formulation of tenofovir disoproxil fumarate (TDF). At the same time, literature data show an onset of dyslipidemia after a switch from TDF to TAF. To better understand the possible role of TAF in dyslipidemia, antiretroviral-naïve HIV-infected patients were evaluated, comparing those treated with TAF/emtricitabine with those with abacavir/lamivudine. **Methods:** We enrolled 270 antiretroviral-naïve HIV-infected patients in an observational, retrospective, longitudinal, multicenter study; they started treatment from 2017 to 2019 and were followed up for at least 72 weeks. We divided patients into two groups, one treated with a TAF-based backbone in their antiretroviral regimens (TAF group) and one without TAF (NO TAF group), to evaluate possible differences in the dynamics of lipid profiles from baseline(T0) to week 24 (T24), 48 (T48) and 72 (T72). **Results:** No significant differences were observed at baseline between the 2 groups. In the TAF group we observed a significant development of hypercholesterolemia throughout the follow-up (*p* < 0.0001), not evident in the NO TAF group, that instead showed a significant increase in high-density lipoprotein (HDL). There were no significant differences between the two groups regarding triglycerides, low-density lipoprotein (LDL) and cardiovascular risk index (CRI). A cholesterol-lowering treatment with statin, finally, was prescribed in 6 patients in both groups during the study. At binary logistic regression analysis, no factor was independently associated with hypercholesterolemia, except for higher age at T0. **Conclusions:** This real-life study shows that in HIV-naïve patients, TAF was associated with hypercholesterolemia throughout the follow-up. The clinical significance of this hypercholesterolemia will have to be clarified in further studies.

## 1. Introduction

Antiretroviral therapy (ART) for human immunodeficiency virus (HIV) infection has considerably changed in recent years. In particular, the introduction of tenofovir alafenamide (TAF) has deeply modified the choice of the backbone, allowing the prevention of bone and renal toxicity related to previous formulations of tenofovir disoproxil fumarate (TDF) [1,2,3,4]. However, some evidence suggests that the switch from TDF to TAF could be associated with a higher incidence of dyslipidemia [5,6,7,8]; this could be due to a “statin-like” effect of TDF, as clinical trials and real-life studies suggest [9]. In some publications, dyslipidemia, after this switch, was associated with higher cardiovascular risk [10]. Thus, TDF results in potential bone and renal toxicity but, at the same time, seems to have an unclear healthy effect in the reduction of plasma lipid levels. In contrast with this observation, other authors suggested that the increase in lipids after switching to TAF was mainly related to the introduction of TAF rather than the end of TDF treatment [11]. Thus, it remains controversial whether the dyslipidemia observed after switching from TDF to TAF is related to the loss of the “TDF statin effect” or to the introduction of TAF. There are few available data in the literature regarding the dynamics of the lipid profile in HIV-naïve patients treated with TAF-based antiretroviral therapy vs. no TAF-based ART.

To better understand whether dyslipidemia is a TAF-related event or a loss of the protective effect of TDF, we evaluated in the present study the dynamics of lipids in the first 72 months of ART among HIV-naïve patients treated with TAF plus emtricitabine (FTC) compared with those treated with abacavir plus lamivudine as a backbone. 

## 2. Patients and Methods

### 2.1. Study Design

We performed an observational, retrospective, longitudinal, multicenter study. We enrolled all the 270 consecutive HIV-naïve outpatients who started treatment in the period from January 2017 until December 2019 and were followed up for at least 72 weeks. Eleven Italian Infectious Disease Centers specialized in HIV treatment (8 in Campania, 2 in Apulia and 1 in Milan) were involved in the study and have cooperated in previous clinical investigations using the same clinical approach and the same laboratory methods.

Inclusion criteria were: (1) a confirmed diagnosis of HIV infection, (2) being naïve to antiretroviral therapy, (3) an age of 18 years or older. Exclusion criteria were: (1) past or present use of lipid-lowering drugs, (2) use of boosted protease inhibitors (bPI) in their antiretroviral regimen that may induce dyslipidemia [12], (3) use of TDF in their antiretroviral regimen that may induce a reduction in lipids [9]; these exclusion criteria were adopted to avoid confounding factors in the interpretation of the final data. The enrolled HIV-naïve patients were divided into 2 groups based on their antiretroviral backbone: a TAF group and a no-TAF group.

HIV infection was classified according to the revised CDC classification system for HIV-infected adolescents and adults, 1993 [13]. Enrolled patients received antiretroviral therapy according to the current international guidelines [14]. Recommendations for statin treatment were in accordance with 2019 ACC/AHA guidelines [15].

All patients underwent clinical examinations and laboratory tests at enrolment, after one month from the beginning of ART treatment, and every 3–6 months subsequently according to the current guidelines [14]. Clinical, immunological, virological and biochemical data were collected and analyzed at enrolment (T0), at 24 weeks (T24), at 48 (T48) and 72 weeks (T72) after beginning ART. 

Being a retrospective real-life study, we considered, as evaluable data, those obtained from patients in a window of about 2 weeks before or after the different timing points of follow-up.

All procedures performed in this study were in accordance with with the 1964 Helsinki declaration and its later amendments or comparable ethics standards. Informed consent was obtained from all participants included in the study.

### 2.2. Study Outcomes

The primary outcome was the evaluation of the development of dyslipidemia at the different time points of the study and its correlation to the backbone of choice in HIV-naïve patients. 

The evaluation of a relationship between an antiretroviral regimen with a metabolic impairment is important, above all when it concerns lipids. There are many data in the literature that identify dyslipidemia as a cardiovascular risk factor [16]. This is even more important for HIV-infected patients who have higher cardiovascular risk (CVR) than HIV-negative individuals [17]. With this in mind, we thought to analyze lipid levels, considering as dyslipidemia the unhealthy levels of one or more kinds of lipid in blood (triglycerides > 150 mg/dl, cholesterol > 200 mg/dl, low-density lipoprotein-LDL ≥ 130 mg/dl) and/or the introduction of statin in every timing of the follow-up. The cut-off of 130 mg/dl for LDL was been chosen on the basis of literature data and is correlated to the absence of CVR for values lower than 130 [18]. This looks more correct above all for HIV-infected patients who have potentially higher underlying CVR [17]. As a secondary outcome, changes in the cardiovascular risk index (CRI) were evaluated at each timepoint to clarify the clinical relevance of the possible lipid increase. CRI was calculated dividing total cholesterol level by HDL level: an increased CRI was considered for values >5 in men and >4.5 in women [19]. Other secondary outcomes were the trend in viro-immunological status (HIV viral load, CD4 and CD8 cell count), the clinically relevant events and other relevant biochemical parameter changes from baseline to week 72. We furthermore analyzed the possible introduction of hypoglycemic drugs in two arms of the study.

### 2.3. Serological Analysis

Antibodies to HIV-1 and -2 were sought using a commercial ELISA (Abbott Lab., North Chicago, IL, USA) and positive results were confirmed by Western blot (Genelabs Diagnostics, Science Park Drive, Singapore). The HIV viral load was assessed by real-time PCR with the lowest detection limit of 40 copies/ml.

Lymphocyte subsets (CD4+, CD8+) were evaluated by flow cytofluorimetry using monoclonal antibodies and a fluorescence-activated cell sorter scan (Becton Dickinson, Mountain View, CA, USA). Serum lipids and routine analyses were performed applying standard procedures. 

### 2.4. Statistical Analysis

Continuous variables were summarized as mean and standard deviation or median and range, and categorical variables as absolute and relative frequencies. For continuous variables, the differences were evaluated by Student’s *t*-test for comparison between two groups and by the one-way analysis of variance (one-way ANOVA) for comparison including 3 or more groups; in the case of non-normally distributed variables the Mann–Whitney U test and the Kruskal–Wallis one-way ANOVA were used for comparison between 2 or more groups, respectively; categorical variables were compared using Fisher’s exact test. Odds ratios (OR) with 95% confidence intervals (CI) were estimated using a binary logistic regression model using the Firth method based on penalized likelihood to identify possible independent associations between hypercholesterolemia and 8 characteristics of the patients enrolled (sex, age, CD4 count, HIV viral load, different antiretroviral regimens). This analysis was applied to each timing of the follow-up. A *p*-value below 0.05 was considered statistically significant.

## 3. Results

The characteristics at enrolment of the 270 HIV-naïve patients included in the study are summarized in Table 1. The median age of patients was 40.5 years (range 18–78), 81.1% were males, 73.7%, Italian subjects, 47.7% men who have sex with men (MSM) and 45.2% had unsafe heterosexual intercourse as risk factor for HIV infection. Thirty-one (11.5%) subjects were at CDC-C stage at enrolment, while the CD4 nadir cell count mean was 359 (±standard deviation, SD: 225) and 40% had a high HIV viral load (>100,000 copies/ml). At enrolment, 21.2% of subjects showed hypertriglyceridemia, 14.1% hypercholesterolemia and 5.3% high serum value of LDL (Table 1).

Of the 270 naïve enrolled patients, 141 were included in the TAF group, and 129 in the NO TAF group. In the TAF group, 79 were treated with TAF/FTC+elvitegravir/cobicistat, 30 with TAF/FTC+dolutegravir, 29 with TAF/FTC+rilpivirine and 3 with TAF/FTC+raltegravir. In the NO TAF group, all 129 subjects enrolled were treated with abacavir/lamivudine+dolutegravir. 

There were no significant differences in the demographic, clinical, viro-immunological or biochemical data at baseline between the 2 groups, except for a higher percentage of late presenters in the Control group (17% vs. 6.3% with *p =* 0.01). All lipid parameters (triglycerides, cholesterol, LDL) were similar in the 2 groups, except for serum HDL which was higher in the TAF group (Table 1).

Analyzing patients for the primary outcome at different times during the follow-up, compared to T0, we observed in the TAF group a significant increase in the prevalence of patients with hypercholesterolemia and/or introduction of statin (11.2% at T0 vs. 30% at T24, 38.4% at T48 and 41.1% at T72; *p* = 0.0000) and at the same time a parallel increase in percentage of LDL ≥ 130 mg/dl (12.5% at T0 vs. 20% at T24, 27.5% at T48 and 27.5% at T72; *p* = 0.03) (Table 2 and Figure 1). In the NO TAF group, no difference was observed in the percentage of patients with hypercholesterolemia and/or the introduction of statin at the different time points (18.5% at T0 vs. 29.4% at T24, 29.2% at T48, 30.7% at T72, *p* = 0.05). There was a progressive increase of pathological LDL values from T0, but unlike with the TAF group, we did not note statistical significance (18.2% at T0 vs. 28% at T24, 26% at T48, 32.6% at T72, *p* = 0.08). In the NO TAF group, there was, instead, an evident significant increase in HDL throughout the follow-up (12.7% at T0 vs. 39.7% at T72, *p* = 0.001) (Table 2 and Figure 1). There were no significant differences between the two groups as regards the prevalence of patients with high triglycerides. As regards CRI, no pathological increase throughout the follow-up was observed in either group.

Finally, we performed a binary logistic regression analysis to evaluate the factors independently associated with hypercholesterolemia, considering sex, age and the two adopted antiretroviral backbone regimens. No factor was independently associated with hypercholesterolemia except at T0 for an age higher than 50 years old (OR: 2.254; 95% CI: 1.175–4.325) (Table 3). 

The same binary logistic regression analysis was performed considering all the different types of treatment in TAF group, screened for different third drug of a regimen, obtaining the same results (Appendix A).

Moreover, all different antiretroviral regimens used for treatment showed the same viro-immunological efficacy with a percentage of HIV suppression at T72, from 90% to 94% (Appendix A). No differences between the TAF and NO TAF groups were observed for lipid averages (Appendix A).

In a supplementary analysis we observed a higher incidence of the introduction of anti-diabetic drugs at different weeks of follow-up in the TAF group vs. the NO TAF group (5.8% vs. 0.9% at T72; *p* = 0.09), although without statistical significance (Appendix A). 

## 4. Discussion

TAF represented a great opportunity for clinicians and, associated to FTC, remains a gold standard as backbone of antiretroviral regimens [14,20]. Contrary to TDF, this drug does not engender bone and renal toxicity and preserves the same virological efficacy [3].

To optimize quality of life and prevent comorbidities (with particular regard to dyslipidemia and cardiovascular complications), many studies have recently examined long-term metabolic disorders. 

The evaluation of relationship between an antiretroviral regimen and a metabolic impairment is important, above all when it concerns lipid alterations. There are many data in the literature that show dyslipidemia as a cardiovascular risk factor [16]. 

Cardiovascular diseases (CVD) represent one of the most important causes of mortality and morbidity in the industrialized world. People with hyperlipemia in fact are at roughly twice the risk of developing CVD as compared to those with normal total cholesterol levels [16]. Patients with familiar hypercholesterolemia (FH) have an even greater risk of developing CVD at an earlier age; therefore, early detection and treatment are imperative to reduce cardiovascular events and premature death [21]. The atherosclerotic process is an important pathogenetic factor of CVD; in particular, hypercholesterolemia is one of most important risk factors that contributes to the formation and progression of atherosclerotic plaque secondary to elevated levels of low-density lipoprotein cholesterol (LDL-C) that initially determine an endothelial dysfunction, an early step in CVD that contributes to plaque initiation and formation [22].

This may explain the wide use of lipid-lowering drugs in general population to avoid hypercholesterolemia [23]. Statins are the mainstay treatment for hyperlipemia; however, the limitations of statins include treatment resistance, intolerance due to adverse events and a lack of adherence, which contribute to poor outcomes [16].

If management of dyslipidemia is so important for prevention of CVD in general population, this is even more crucial in patients with HIV infection. People living with HIV (PLWHIV) in fact have a 2-fold higher risk of having a cardiovascular event than HIV-negative individuals [17]. On the other hand, HIV-infected patients have to use lifelong antiretroviral drugs that have to be optimized to reduce long-term metabolic impact. Regarding antiretroviral therapy (ART), there are many data in the literature on dyslipidemia associated with a switch from TDF to TAF in treatment-experienced HIV patients [5,6,7,8]. In particular, in a recent real-world study on 6423 HIV-experienced patients, Brunet et al. showed a development of dyslipidemia associated however to an underutilization of statin after switching from TDF to TAF [24]. More recently clinicians have evaluated not only the metabolic aspects but also a possible onset of weight gain in patients who showed dyslipidemia after antiretroviral treatment. A recent study on a Swiss HIV cohort showed development of weight gain, obesity and a worsening in serum lipid levels after switching from TDF to TAF [25]. Similar conclusions appear in a Chinese study, which observed an increase in dyslipidemia associated with weight gain in patients switching to TAF/FTC/EVG/COBI [26]. In a Spanish study, the authors analyzed this aspect no longer in a switch strategy, but comparing naïve patients who started the same antiretroviral regimen, but with a different NRTI backbone. The authors concluded that there was a worse lipid profile and greater statin use in patients treated with TAF/FTC plus Elvitegravir/cobicistat (EVG/COBI) than those treated with TDF/FTC plus EVG/COBI [11]. A recent Italian work confirmed the onset of dyslipidemia after switching from TDF/FTC to TAF/FTC, also associated with greater glycemia [27]. This aspect seems to be confirmed also in patients without HIV infection, but treated with TAF for HBV infection. A recent work of Suzuki et al. confirms, in fact, the onset of dyslipidemia after switching from TDF to TAF in a cohort of patients treated for HBV hepatitis, without HIV infection [28]. Another interesting work showed that an increase in lipids was a reversible effect that appeared when switching from TDF to TAF and disappeared if the patients were re-treated with TDF [29]. Other authors evaluated the role of a third drug associated with TAF in the evolution of dyslipidemia. Some data, for example, seem to suggest a possible protective effect of rilpivirine in lipidic dynamics during TAF/FTC treatment, as evident in an Italian paper. In this real-life study, after switching from TDF/FTC+rilpivirine to TAF/FTC+rilpivirine, the authors observed a slight increase in lipids except for patients with hypercholesterolemia at baseline [30].

These studies enrolled mostly experienced patients and so it is not completely clear if the increase in lipid levels was a side effect of TAF, if it was related to TDF discontinuation, or if it was due to both factors, as suggested by other studies. In Sword 1 and 2 studies, in particular, patients who were efficaciously treated with a standard triple regimen (with prevalently TDF in NRTI backbone) were switched to dual regimens based on the association of dolutegravir/rilpivirine. After the switch, in spite of TDF discontinuation, the patients showed a good profile of tolerability without an onset of dyslipidemia [31]. On the other hand, there are also some data that seem to refute an association between switching from TDF to TAF and dyslipidemia. Schwarze-Zander et al. evaluating 347 PLWH who underwent a switch from TDF to TAF observed a reduction in renal toxicity without an increase in the lipid profile [32]. Thus, it remains controversial whether dyslipidemia observed after switching from TDF to TAF is related to the loss of “TDF statin effect” or to the introduction of TAF. 

In conclusion there are few available data in the literature regarding the dynamics of lipid profile, evaluated in clinical practice, in HIV-naïve patients treated with TAF-based ART vs. no TAF-based ART. 

To better understand the direct impact of TAF in the evolution of dyslipidemia, we designed the present study enrolling a group of HIV-infected patients, naïve to antiretroviral treatment and, thus, potentially free from any confounding factors. 

Our preliminary real-life data showed that in naïve patients, TAF was associated with a higher percentage of development of hypercholesterolemia throughout the follow-up, although not confirmed at binary logistic regression analysis. Instead, in the subjects treated with NO TAF-based ART, the development of hypercholesterolemia was not statistically evident, being associated only with a significant HDL increase, with a potential protective effect exerted by this fraction of cholesterol. In any case, in both groups no changes in CRI levels were observed. 

This study has some limitations, i.e., having a retrospective design. Moreover, the study was concluded in December 2019, consequently, no patient had been treated with bictegravir because this drug was not available at that time. However, to the best of our knowledge, there are few data in the literature on the lipid profile in antiretroviral-naïve HIV-infected patients, treated in real-life with TAF vs no TAF. 

In conclusion, our study, based on data of real-life, highlights that the use of TAF-based regimen in ART-naïve HIV-infected patients, seems to be correlated to dyslipidemia along the follow-up. This may induce a higher CVR in this group of patients because the elevation of total cholesterol and LDL correlate with the development of atheromatous plaques and so with CVD, as evident in literature [23]. Our data may so supply interesting elements to optimize the management of ART regimens in HIV-infected patients with CVR and/or weight gain.

Further observations on a larger number of patients and perhaps the contemporary collection of data obtained with the Doppler of the supra-aortic trunks, will be able to confirm our observations and will help to better define CVR in this setting.

## Figures and Tables

**Figure 1 biomedicines-10-03164-f001:**
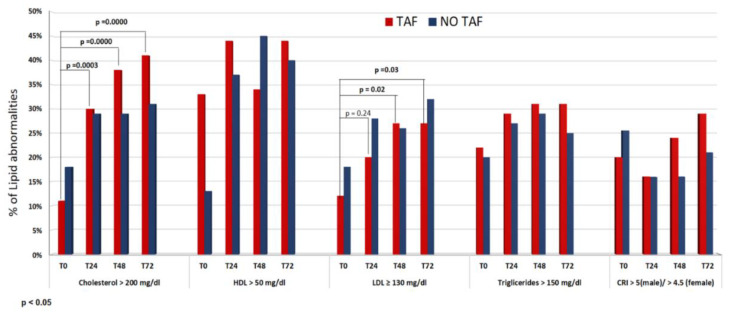
Percentage of lipid abnormalities and CRI at different timing of follow-up according to ARV backbone.

**Table 1 biomedicines-10-03164-t001:** Demographic, clinical, viro-immunological and biochemical data of the 270 patients at enrolment according to ARV backbone.

BASELINE DATA	TOTAL	TAF GROUP	NO TAF GROUP	*p*
Numbers of patients	270	141	129	
Age median (range)	40.49 (18–78)	39 (19–74)	40 (18–78)	
Male/Female	219/51	114/27	105/24	
Italians/Strangers	199/71	100/41	99/30	
Risk Factors for HIV infection	MSM 129 (47.77%) PWID 19 (7.03%) HETERO 122 (45.18%) OTHER 4 (1.48%)	MSM 71 (50.35%) PWID 11 (7.80%) HETERO 61 (43.26%) OTHER 2 (1.41%)	MSM 58 (44.96%) PWID 8 (6.20%) HETERO 61 (47.28%) OTHER 2 (1.55%)	
Antiretroviral Regimens	TAF–based 141 (52%) NO TAF–based 129 (48%)	TAF+FTC+RIL 29 (20%) TAF+FTC+EVG/c 79 (56%) TAF+FTC+DGT 30 (21%) TAF+FTC+RAL 3 (2%)	ABC+LAM+DGT 129 (100%)	
CDC CLASSIFICATION: C (late presenters)	C: 31 (11.48%)	C: 9 (6.38%)	C: 22 (17%)	**0.01**
CD4+ NADIR, cell/µL mean ± SD	359.54 ± 225.83	380.4 ± 221.22	336.38 ± 229.49	0.10
HIV-RNA > 100.000 copies/ml	108 (40%)	56 (39.71%)	52 (40.31%)	0.98
Triglycerides (mg/dl) mean± SD	112.82 ± 60.22	111.60 ± 63.27	114.31 ± 59.49	0.71
Cholesterol mean (mg/dl) ± SD	156.96 ± 40.99	154.51 ± 39.78	160.01 ± 42.43	0.27
HDL mean (mg/dl) ± SD	44.07 ± 21.50	47.30 ± 25.22	40.30 ± 15.36	**0.006**
LDL mean (mg/dl) ± SD	97.95 ± 34.31	97.12 ± 31.29	98.86 ± 37.49	0.67
% of Elevated Triglycerides > 150 mg/dl	51/240 (21.25%)	29/132 (21.96%)	22/108 (20.37%)	0.88
% of Elevated Cholesterol > 200 mg/dl	34/241 (14.10%)	14/133 (10.52%)	20/108 (18.51%)	0.11
% of Elevated HDL > 50 mg/dl	48/2023 (23.64%)	36/109 (33.02%)	12/94 (12.76%)	**0.001**
% of Elevated LDL ≥ 130 mg/dl	9/170 (5.29%)	11/88 (12.5%)	15/82 (18.29%)	0.40

MSM, Men who have sex with men; PWID, Persons Who Inject Drugs; HETERO, Heterosexuals. TAF, Tenofovir alafenamide; FTC, Emtricitabine; RIL, Rilpivirine; EVG/COBI, Elvitegravir/Cobicistat; DGT, Dolutegravir; RAL, Raltegravir; ABC, Abacavir; LAM, Lamivudine. CDC, Centers for Disease Control.

**Table 2 biomedicines-10-03164-t002:** Lipidic dynamics and CRI at different times of follow-up according to ARV backbone.

	TAF T0	TAF T24	*p* T24 vs. T0	TAF T48	*p* T48 vs. T0	TAF T72	*p* T72 vs. T0
% of Elevated Triglycerides >150 mg/dl	29/132 (21.96%)	39/132 (29.54%)	0.20	32/104 (30.76%)	0.16	32/102 (31.37%)	0.14
% of Elevated Cholesterol > 200 mg/dl and/or statin use	15/133(11.27%)	40/133 (30%)	**0.0003**	40/104 (38.46%)	**0.0000**	42/102 (41.17%)	**0.0000**
Patients treated by statine at different follow-up periods	0	5 (3.7%)	0.21	6 (5.7%)	0.06	5 * (4.9%) * 1 patient with statin at T48 lost at follow-up	0.056
Patients with elevated Cholesterol Patients with statin and normal cholesterol Patients with statin and elevated cholesterol	14 1 0	38 2 3		39 1 5		39 3 2	
% of Elevated HDL ≥ 50 mg/dl	36/109 (33.02%)	51/116 (43.96%)	0.12	28/82 (34.14%)	0.99	34/77 (44.15%)	0.16
% of Elevated LDL ≥ 130 mg/dl	11/88 (12.5%)	19/95 (20%)	0.24	19/69 (27.53%)	**0.02**	16/58 (27.58%)	**0.03**
% CRI [>5 (Male)/>4.5 (Female)]	22/109 (20.18%)	19/116 (16.37%)	0.57	20/82 (24.39%)	0.6	22/77 (28.57%)	0.24

	NO TAF T0	NO TAF T24	*p* T24 vs. T0	NO TAF T48	*p* T48 vs. T0	NO TAF T72	*p* T72 vs. T0
% of Elevated Triglycerides > 150 mg/dl	22/108 (20.37%)	27/100 (27%)	0.33	28/96 (29.16%)	0.19	28/110 (25.45%)	0.46
% of Elevated Cholesterol > 200 mg/dl and/or statin use	20/108 (18.51%)	30/102 (29.41%)	0.09	29/99 (29.29%)	0.09	35/114 (30.7%)	0.05
Patients treated by statine at different follow-up periods	0	0		6 (6%)		5 * (4.3%) * 1 patient with statin at T48 lost at follow-up	
Patients with elevated Cholesterol Patients with statin and normal cholesterol Patients with statin and elevated cholesterol	20 0 0	30 0 0		29 0 6		32 3 2	
% of Elevated HDL ≥ 50 mg/dl	12/94 (12.76%)	26/70 (37.14%)	**0.0005**	30/67 (44.77%)	**0.000**	31/78 (39.74%)	**0.0001**
% of Elevated LDL ≥ 130 mg/dl	15/82 (18.29%)	16/57 (28%)	0.24	13/50 (26%)	0.4	17/52 (32.6%)	0.08
% CRI [>5 (Male)/> 4.5 (Female)]	24/94 (25.53%)	11/70 (15.71%)	0.18	11/67 (16.41%)	0.23	16/78 (20.51%)	0.55

CRI: Cardiovascular Risk Index.

**Table 3 biomedicines-10-03164-t003:** Binary logistic regression analysis evaluating the independent factors associated to hypercholesterolemia at different time points.

Binary Logistic Regression				
**Follow-up 24 Weeks**	**Sig.**	**Exp (B)**	**95% CI for EXP (B)**	
			**Inferior**	**Superior**
Age ≥ 50 vs. < 50 years old	**0.015**	2.254	1.175	4.325
Sex F vs. M	0.104	1.745	0.892	3.413
TAF vs. NO TAF	0.991	0.997	0.558	1.781
**Follow-up 48 weeks**	**Sig.**	**Exp (B)**	**95% CI for EXP (B)**	
			**Inferior**	**Superior**
Age ≥ 50 vs. < 50 years old	0.202	1.545	0.792	3.012
Sex F vs. M	0.663	1.169	0.579	2.361
TAF vs. NO TAF	0.204	1.466	0.812	2.647
**Follow-up 72 weeks**	**Sig.**	**Exp (B)**	**95% CI for EXP (B)**	
			**Inferior**	**Superior**
Age ≥ 50 vs. < 50 years old	0.25	1.495	0.748	2.987
Sex F vs. M	0.67	0.531	0..270	1.047
TAF vs. NO TAF	0.405	0.783	0.440	1.3393

TAF, Tenofovir alafenamide.

## Data Availability

The data presented in this study are available on request from the corresponding author.

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
