# Peer review of "Dynamics of Lipid Profile in Antiretroviral-Naïve HIV-Infected Patients, Treated with TAF-Based Regimens: A Multicenter Observational Study"

_biomedicines, 2022, doi:10.3390/biomedicines10123164_

Round 1

Reviewer 1 Report

The manuscript by Martini and colleagues presents the outcome of the lipid profile in real-life cohorts of treatment-naive PLWH on cART with the backbone of TAF/FTC and ABC/3TC. The authors monitored the dynamics of the lipid profile at baseline, weeks 24, 48, and 72. The results showed that the TAF group had a significant development of 38 hypercholesterolemia throughout the follow-up. It is not clear what is the clinical significance of hypercholesterolemia remains unknown. This study appears to be an afterthought or residual of some other study. While neither of these two factors demotes the value of the study, the clinical significance of the study remains negligible. There are many errors in the manuscript. For example, the last paragraph of the 'Introduction' section defines PLWH as HIV-naive people living with HIV. Is this something new? How HIV naive people can live with HIV? Similarly, line 85 states that 'these two' exclusion criteria. In fact, there are three criteria instead. I am puzzled why some sections are highlighted. 

Author Response

  1. Point 1: It is not clear what is the clinical significance of hypercholesterolemia remains unknown.

Clinical significance of hypercholesterolemia is based on its correlation with higher cardiovascular risk justifying the use of statins not only in HIV patients but also in general population. We furthermore reported in reference the significance of hypercholesterolemia in HIV-infected patients (Maggi P. et al. Cardiovascular risk and dyslipidemia among persons living with HIV: a review. BMC Infect Dis. 2017). In every case we added a sentence, in the “study outcome”, to better describe the clinical importance of the target of our study. (It is known that dyslipidemia is a risk factor for development of athermatous plaques and for cardiovascular disease. This is evident in general population, not only in HIV patients justifying the wide use of lipid-lowering drugs).

  1. Point 2: This study appears to be an afterthought or residual of some other study.

In our study we have evaluated in real-life a multicenter cohort of HIV-infected patients, naïve to antiretroviral therapy. Most of previous data, reported also in references, were derived from trials designed to evaluate non-inferiority of the TAF vs no TAF regimen, rather than focused on metabolic parameters. Most of data, moreover, were related to switch strategy from TDF to TAF. In our study instead there are not confounding factors, because we evaluate metabolic parameters in 2 groups of HIV naïve patients who started treatment (TAF vs ABC).

  1. Point 3: While neither of these two factors demotes the value of the study, the clinical significance of the study remains negligible.

About clinical significance, we thought that knowing whether or not the TAF-based regimen is related to dyslipidemia may help clinicians to optimize the choice of treatment in HIV-infected patients with metabolic syndrome and/or higher potential cardiovascular risk.

  1. Point 4: There are many errors in the manuscript. For example, the last paragraph of the 'Introduction' section defines PLWH as HIV-naive people living with HIV. Is this something new? How HIV naive people can live with HIV?

As suggested by reviewer, we have corrected definition in the text and in the title of the paper changing PLWH in HIV-naïve patients.

  1. Point 5: Similarly, line 85 states that 'these two' exclusion criteria. In fact, there are three criteria instead.

About exclusion criteria, we wanted to indicate “two” meaning the last two cited criteria, related to excluded drugs (bPI and TDF). In every case we have corrected the sentence to make it clearer.

Reviewer 2 Report

There are few available data in the literature regarding the dynamics of lipid profile, evaluated in clinical practice, in HIV-naïve patients treated with TAF-based antiretroviral therapy vs. no TAF-based ART. To better understand the direct impact of TAF in the evolution of dyslipidemia, we designed the present study enrolling a group of PLWH, naïve to antiretroviral treatment and, thus, potentially free from any confounding factors. Our preliminary real-life data showed that in naive PLWH, TAF was associated with a higher percentage of development of hypercholesterolemia throughout the follow-up, although not confirmed at binary logistic regression analysis.

the study may supply interesting data to optimize the management of antiretroviral-naïve PLWH, especially for those with cardiovascular risk and/or weight gain. 

The article is of interest to open other, more specific lines of research. Therefore, it is worth being accepted.

Author Response

The article is of interest to open other, more specific lines of research. Therefore, it is worth being accepted.

Response: Thanks to the reviewer for the appreciation of the study.

Reviewer 3 Report

The work described in this article is interesting and well analyzed. The only substantial observation to the method is that, having to compare the lipid profile of two groups, I would have chosen to compose them in a homogeneous way from the lipid point of view, while in this work the group of patients undergoing treatment with TAF has higher serum HDL levels at baseline. Otherwise the work is correct. I just add a few form observations which I hope will help to further improve the article

the first sentence in the abstract is a bit cryptic.

use of a boosted protease inhibit000000or please check

if one Inclusion criteria was being naïve to antiretroviral therapy, why are there exclusion criteria such as “use of a boosted protease inhibitor (bPI) in their first antiretroviral regimen that may induce dyslipidemia [12], 3) use of TDF in their first antiretroviral regimen”

in according to please check

decimals are sometimes indicated with a comma and sometimes with a point

pag. 5, line 5: please specify that figure 1 is supplementary material

the format of the citations is to be reviewed

Author Response

Response to Reviewer 2 Comments

Point 1: the first sentence in the abstract is a bit cryptic.

Response 1: Thanks to the reviewer for the suggestions. I have modified the first sentence in the abstract to make it more clear.          (Please see corrected paper in the attachment ).

Point 2: use of a boosted protease inhibit000000orplease check

Response 2: It has been an error of transcription, I have corrected the word writing “inhibitor”.

Point 3: if one Inclusion criteria was being naïve to antiretroviral therapy, why are there exclusion criteria such as “use of a boosted protease inhibitor (bPI) in their first antiretroviral regimenthat may induce dyslipidemia [12], 3) use of TDF in their first antiretroviral regimen”

Response 3: The sentence was related to patients who were retrospectively analyzed, however to avoid a wrong interpretation, I have eliminated the word “first”.

Point 4: decimals are sometimes indicated with a comma and sometimes with a point

Response 4: I have corrected numbers in tables and figures, using points to indicate decimals.                                                                    (Please see corrected paper in the attachment)

Point 5: 5, line 5: please specify that figure 1 is supplementary material

Response 5: Figure 1 was not in supplementary material, therefore I have inserted it in the text, because was related to principal data of the study.  (Please see corrected paper in the attachment ).

Point 6: the format of the citations is to be reviewed

Response 6: I have corrected the format of the citations, as suggested. (Please see corrected paper in the attachment ).

Round 2

Reviewer 1 Report

There are still problems with the revisions. In addition to responding to the comments directly to the reviewer, the authors should include the relevance of comments in the manuscript. Agreeing or disagreeing with the reviewer does not enhance the impact (or science) of the manuscript. Whatever is not agreed upon should be discussed in the manuscript to justify the response.

Author Response

(The authors gave the same response as above.)
